PREPARED FOR SUBMISSION TO JHEP

# Apparent fine tunings for field theories with broken space-time symmetries

**Alberto Nicolis,**[1] **Ira Z. Rothstein**[2]

[1] *Center for Theoretical Physics and Physics Department,*
*Columbia University, New York, NY 10027, USA*

[2] *Dept. of Physics, Carnegie Mellon University, Pittsburgh PA 15213, USA*

ABSTRACT: We exhibit a class of effective field theories that have hierarchically small Wilson coefficients for operators that are not protected by symmetries but are not finely tuned. These theories possess bounded target spaces and vacua that break space-time symmetries. We give a physical interpretation of these theories as generalized solids with open boundary conditions. We show that these theories realize unusual RG flows where higher dimensional (seemingly irrelevant) operators become relevant even at weak coupling. Finally, we present an example of a field theory whose vacuum energy relaxes to a hierarchically small value compared to the UV cut-off.

## 1   Introduction

For a generic quantum field theory, the renormalization program asserts that all operators that are permitted by symmetries should be included in the action. Given a set of operators, one defines two types of fine tunings: i) Couplings that are not radiatively stable, yet are smaller than their natural values (based upon dimensional analysis), like the Higgs mass or the cosmological constant. ii) Those that are protected by symmetry, but which take on smaller values than expected from dimensional analysis, like the electron mass. The former are considered the more egregious of the two fine-tunings.

     The fine tuning of the electro-weak scale has driven the model-building community for the past forty years. Sensible extensions of the standard model have been built that lead to a UV symmetry enhancement above the weak scale (e.g. supersymmetry), which in turn resolves the electroweak hierarchy issue. Other approaches [1] involve theories that have no dimension-two scalar operators in the spectrum (Technicolor), while still others invoke a logarithmic running of the relevant Higgs mass operator from the high scale to the low scale via strong coupling (warped extra dimensions). In all of these cases new physics is required at scales not too much above the electro-weak one and, at present, we have seen no solid evidence of physics beyond the standard model. Building models to solve the cosmological constant (CC) problem—an even more egregious fine tuning—is more challenging since it requires new physics at longer distance scales.

An alternative class of solutions to these problems use dynamical relaxation mechanisms. The idea behind this approach is based on the notion that if couplings are thought of as expectation values of some dynamical fields, then their low energy renormalized values will be driven, by energy minimization considerations, to be small compared to their natural values. Such an approach was used by Peccei and Quinn [2] to explain the vanishing of the QCD theta parameter. This apparent fine-tuning is distinct from the hierarchy or CC problems in that it involves a tuning of a dimensionless parameter, $\theta$, the coupling for $F\tilde{F}$ in QCD. The basic idea behind the Peccei-Quinn (PC) mechanism is that $\theta$ gets promoted to the status of a zero mode of a Goldstone boson that couples to the divergence of the Chern-Simons current. The associated $U(1)$ symmetry is broken explicitly (by instantons), which leads to a potential for the Goldstone that drives the expectation value to zero. This idea was generalized to a dynamical, cosmological, setting by Abbott [3], who used it to build a model whose cosmological constant relaxes to zero or some small number. More recently cosmological relaxation mechanisms have been utilized in the context of the standard model hierarchy [4].

The aim of this paper is to show that, if we relax some assumptions and consider a larger space of field theories, then there is a class of theories where a relaxation mechanism for Wilson coefficients naturally arise. Moreover, there is no need to invoke a new set of degrees of freedom or to even worry about generating potentials for moduli as one does in the case of the PQ mechanism. We will see that relaxation can occur generically when we bound the target space of the theory.

## 1.1 Lessons from all around us

It is interesting to note the if we consider non-fundamental theories, we find that apparent fine-tunings abound. For instance, consider any macroscopic object, where the UV cutoff is of order of the atomic spacing. If we allow for open boundary conditions (i.e. no external stresses), the internal stresses relax to zero, or very small values—$T_{ij} \ll \Lambda$, where $\Lambda$ is the UV cut-off in the appropriate units. This constraint will naturally lead to a set of vanishing, or near vanishing, couplings for the effective field theory that describes excitations about this relaxed state, with no recourse to any symmetries, as we will demonstrate below, thus violating i). As we shall see, for macroscopic objects these couplings are suppressed, in cutoff units, by the extrinsic curvature of the boundary of the object under consideration. Thus, we cannot attribute this suppression solely to the open boundary conditions, but to the fact that the ground state has evolved over a period of time, say by accruing matter, such that the radius of our object is much larger than the atomic spacing. In the language of our field theory this will correspond to a hierarchy in the scales appearing in the local action and the scale set by the boundary of the target space.

*Conventions:* We work in natural units ($\hbar = c = 1$) and use the mostly-plus signature for the metric throughout the paper.

## 2 The relaxed 3D solid

We begin by considering the effective theory of a solid [5–7]. In the continuum limit, the dynamical degrees of freedom are the comoving coordinate labels of the solid elements $\phi^I(x,t)$, which, from the viewpoint of spacetime symmetries, are just three scalar fields.

We usually consider solids that admit homogeneous ground states—again, in the continuum limit. This corresponds to choosing the comoving coordinates to be aligned with the spatial ones on such homogeneous ground states, $\langle \phi^I \rangle \propto x^I$, and to impose a shift (internal translation) invariance $\phi^I \to \phi^I + a^I$ on the action. In this way, the ground state corresponds to a configuration that breaks the shift symmetry along with spatial translations down to the diagonal subgroup.

Notice that the shift symmetry implies that there is at least one derivative per field. Furthermore, the ground state field configuration has nonzero first-derivatives. These two properties suggest that we should choose our power counting such that single derivatives of the fields are order-one while higher derivatives are suppressed. The most general action to lowest order then will take the form

$$S = \int G(B^{IJ}) \, d^4x \ , \tag{2.1}$$

where

$$B^{IJ} = \partial_\mu \phi^I \partial^\mu \phi^J \ , \tag{2.2}$$

and $G$ is a generic function, related to the solid's equation of state [1].

If the solid's ground state features also some rotational symmetry group $H \supset SO(3)$, such as the cubic group, this should be imposed as an internal symmetry acting directly onto the $\phi^I$ fields, thereby restricting the form of $G$. The ground state $\langle \phi^I \rangle \propto x^I$ will then break the internal $H$ and spatial rotations down the diagonal $H$ subgroup. For simplicity, below we will consider isotropic solids only, so that there is an internal $SO(3)$ acting on our fields, but much of what we say applies to anisotropic solids as well.

Notice that the action (2.1), regardless of the actual $G$ we choose, always admits generic *physically homogeneous* solutions of the form

$$\langle \phi^I \rangle = \alpha^I{}_\mu x^\mu \ , \tag{2.3}$$

for arbitrary constant $\alpha^I{}_\mu$. Despite formally breaking spacetime translations, these solutions are homogeneous in that physical quantities such as the stress-energy tensor and the currents associated with the internal symmetries are constant in spacetime. This is what we mean when we talk about homogeneous solids.

A nonzero $\alpha^I{}_0$ corresponds to a nontrivial speed for our comoving coordinates, that is, to a moving solid. Let us choose a frame where the solid is at rest. We are left with

$$\langle \phi^I \rangle = \alpha^I{}_j x^j \ , \tag{2.4}$$

where the $3 \times 3$ matrix $\alpha^I{}_j$, since it belongs to $GL(3, \mathbb{R})$, can be written as the composition of a rotation, a shear, and a dilation (see for instance [14]).

We will be considering ground states with no shear. As to the rotation part, it can be undone by re-orienting the spatial axes, similarly to the choice of frame we made above. We are thus left with a single dilation parameter $\alpha$, which can be thought of as the compression level—or scale factor—of our solid:

$$\langle \phi^I \rangle = \alpha \, \delta^I_j x^j \ . \tag{2.5}$$

---

[1]This action is also derivable using the coset construction [8].

By changing the external pressure, or tension, we change the value of $\alpha$. Releasing the external pressure and letting go of the boundaries of the solid, we let $\alpha$ relax to a particular value, determined by $G$, as we now show.

Notice that, for a solid, the boundary can move and deform in physical space, but is fixed in comoving space. So, we can parametrize the boundary of our solid through a constraint equation of the form $F(\phi) = 0$, and the interior through $F(\phi) > 0$, for some function $F$. For instance, for a spherical solid, we can choose $F$ to be $R^2 - \phi^I \phi^I$, where $R$ is the comoving radius of the solid.

Now, how are we to alter the solid action (2.1) to take into account that the solid has a fixed boundary in comoving space and that the boundary is free to move in physical space? The most convenient approach seems to be to write the action as [13]

$$S = \int \theta\big(F(\phi)\big) G(B^{IJ}) d^4 x \, , \tag{2.6}$$

and leave the variational principle untouched: the $\theta$-function ensures that only the interior of the solid contributes to the action, and the fact that no constraint on the field variations is imposed at the boundary of the solid ensures that the fields are free to vary there, that is, that the boundary is free to move.

One can derive the equations of motion just by varying the action above; however, for our purposes, it is more convenient to phrase them in terms of stress-energy conservation. The stress-energy tensor reads

$$T_{\mu\nu} = \theta\big(F(\phi)\big)\tilde{T}_{\mu\nu} \, , \tag{2.7}$$

where $\tilde{T}_{\mu\nu} = -\frac{2}{\sqrt{g}}\frac{\partial(\sqrt{g}\,G)}{\partial g^{\mu\nu}}$ and we have used the fact that $F(\phi)$ is independent of the space-time metric. Stress-energy conservation for a static configuration then implies

$$\partial_i T^{i\nu} = \delta\big(F(\phi)\big)\partial_i F(\phi)\,\tilde{T}^{i\nu} + \theta\big(F(\phi)\big)\partial_i\tilde{T}^{i\nu} = 0 \, . \tag{2.8}$$

The coefficient of the $\theta$-function gives us the usual bulk eom's for a static system,

$$\partial_i \tilde{T}^{i\nu} = 0 \, . \tag{2.9}$$

In addition, however, we have boundary equations of motion coming from the $\delta$-function. Considering a ground state of the form (2.5) and using the fact that $n_i(\phi^\star) \propto \partial_i F(\phi)\big|_{\phi^\star}$ is the normal to the boundary at the boundary point $\phi^I = \phi^{I\star}$ [2] we have

$$n_i(\phi^\star)\,\tilde{T}^{i\nu}(\phi^\star) = 0 \, . \tag{2.10}$$

Given that $\tilde{T}_{\mu\nu}$ in the ground state (2.5) is homogenous and that in moving around a closed boundary we let $n_I(\phi^\star)$ explore all possible spatial directions, we then have, in particular,

$$\tilde{T}^{ij} = 0 \quad \text{everywhere.} \tag{2.11}$$

This is just the relaxation of the tension of the solid when the boundaries are free to adjust.

Notice that for a configuration like (2.5), the stress-energy tensor depends on $\alpha$. So, eq. (2.11) can be obeyed only if there is a value of $\alpha$ for which a certain, $G$-dependent condition is obeyed.

---

[2] We will drop the $I$ index on $\phi^{\star I}$ from here on to avoid clutter and confusion with indices.

This is akin to minimizing a potential: if the potential has no minimum, there are no static solutions. Specifically, the bulk stress-energy tensor reads

$$\tilde{T}_{\mu\nu} = -2 \frac{\partial G}{\partial B^{IJ}} \partial_\mu \phi^I \partial_\nu \phi^J + \eta_{\mu\nu} G = 0 \ . \tag{2.12}$$

On the ground state (2.5) we have $B^{IJ} = \alpha^2 \delta^{IJ}$, and, because of isotropy, $\frac{\partial G}{\partial B^{IJ}} = A(\alpha)\delta_{IJ}$, for some $A(\alpha)$. Then, calling $G(\alpha)$ the value of $G$ evaluated on the ground state, we have $G'(\alpha) = 6\alpha A(\alpha)$. So, the relaxation condition (2.11) can be rewritten as

$$3\,G(\alpha) - G'(\alpha)\,\alpha = 0 \ . \tag{2.13}$$

For any given solid, its $G$ function is fixed. So, this equation should not be interpreted as a differential equation for $G$, but rather as an algebraic equation for $\alpha$. As we mentioned, it should be thought of as the analog of finding the minimum of a potential. Indeed, if we plug the static ansatz (2.5) directly into our action (2.6), we get an effective potential for $\alpha$: [3]

$$V_{\text{eff}}(\alpha) = -\mathcal{V}\,G(\alpha)\alpha^{-3} \ , \tag{2.14}$$

where $\mathcal{V} \equiv \int d^3\phi\,\theta(F(\phi))$ is the *comoving* volume of the solid, which is a constant. Minimizing $V_{\text{eff}}$ w.r.t. $\alpha$ is equivalent to imposing our relaxation condition (2.13).

How generic is the existence of a solution to eq. (2.13)? *All* solids that exist in nature at zero (or negligible) external pressures find that special value of $\alpha$ and call that their ground state. For others, that special value might not exist: for instance, at zero temperature solid helium only exists at pressures above $\sim 25$ atm, and it melts if the pressure drops to lower values.

Consider now the dynamics of excitations about the ground state,

$$\phi^I(x) = \alpha\big(x^I + \pi^I(x)\big) \ . \tag{2.15}$$

If we expand the bulk action, we find

$$S_{\text{bulk}} = \int d^4x \left( G_0 + \frac{\partial G}{\partial B^{IJ}}\bigg|_0 s^{IJ} + \frac{1}{2} \frac{\partial G}{\partial B^{IJ} \partial B^{KL}}\bigg|_0 s^{IJ} s^{KL} + \dots \right) \ , \tag{2.16}$$

where $s^{IK}$ is the generalized strain defined as

$$s^{IK} = (\partial_\mu \phi^I \partial^\mu \phi^K - \alpha^2 \delta^{IK}) = \alpha^2 \big(2\,\partial^{(I} \pi^{K)} + \partial_\mu \pi^I \partial^\mu \pi^K\big) \ , \tag{2.17}$$

and the subscript zero reminds us to evaluate $G$ and its derivatives on the ground state. The symmetries of the original bulk action and of the background imply certain relationships among the Wilson coefficients of the expanded action. However, in general the relaxation condition (2.13) will imply *additional* relationships, which are not just a consequence of symmetry. These will look like fine tunings.

---

[3]In general, the reduced variational problem one gets by plugging an ansatz into an action is equivalent to the original one only if certain symmetry-based conditions are obeyed—see e.g. [14, 15]. From this viewpoint, this procedure is not entirely warranted in our case, since we have not allowed for the most general ansatz given the broken symmetries.

For the 3D solid we are considering, these constraints lead to the statement that the free energy is independent of the anti-symmetric part of $\partial_i \pi_J$. Physically this is the statement of the invariance of the free energy under under rotations up to quadratic order. This is discussed in Appendix A. As we will now see, upon dimensional reduction (see Appendix B), the open boundary conditions lead to the vanishing of a *relevant* operator.

## 3 The oscillating bar

As our simplest and perhaps most surprising example we consider a one-dimensional embedded solid—a bar. Consider the action for the transverse oscillations $\vec{\varphi}(x, t)$ of such a bar with at least one open end,

$$S = \int dx dt \frac{1}{2} (\dot{\vec{\varphi}}^2 - c_1 \vec{\varphi}''^2) + \dots \tag{3.1}$$

where $c_1$ is a bar-dependent (dimensionful) coefficient, primes denote derivatives with respect to $x$, and the $\dots$ includes non-linear derivatively coupled interactions (see e.g. [9]). The glaring absence of a standard two-derivative gradient energy term, $\vec{\varphi}'^2$, would appear to be a fine tuning as there exists no symmetry explanation[4]. On the other hand, this well known property was discovered by Euler and Bernoulli who derived the equations of motion (the "beam equation") from microscopic considerations [9]. Eq. (3.1) implies, in particular, that the normal oscillation frequencies of the bar are inversely proportional to the *square* of the bar's length[5]. Apparently, the IR renormalized coupling of $\vec{\varphi}'^2$ relaxes to zero independently of the composition of the bar, i.e. the UV physics.

Let us then consider the effective theory description of this system to understand how this apparent fine tuning comes about. Unlike for a fundamental string, we can label matter elements along a material string or bar by a real number $\phi(\sigma, \tau)$, where $\sigma$ and $\tau$ are generic worldsheet coordinates. This has to do with the spontaneous breaking of spacetime symmetries *along* the string or bar, and is explained at length in ref. [16]. To derive an action we may extend the formalism of the previous section to a one-dimensional solid, with one open end corresponding to, say, $\phi = \phi^\star$, embedded in a four-dimensional Minkowski space, parametrized by embedding coordinates $X^\mu(\sigma, \tau)$. In particular, the induced metric on our solid's worldsheet is $g_{\alpha\beta} = \partial_\alpha X^\mu \partial_\beta X_\mu$, with $y^\alpha = (\sigma, \tau)$. Thus, to lowest order in derivatives the action is the generalization of the Nambu-Goto action

$$S_{\text{bar}} = \int d\tau d\sigma \, \theta(\phi - \phi^\star) \sqrt{g} \, G(B) , \qquad B \equiv g^{\alpha\beta} \partial_\alpha \phi \partial_\beta \phi , \tag{3.2}$$

where, as before, $G$ is an arbitrary function.

The action is reparametrization invariant, and it is convenient to choose "unitary" gauge,

$$X^0 = \tau \equiv t , \qquad X^1 = \sigma \equiv x , \tag{3.3}$$

---

[4]Note that the interaction terms left off in the action have one derivative per field, which are invariant only under a shift symmetry.

[5]We thank Ben Freivogel and Federico Piazza for educating us regarding this. It is thanks to conversations with them that this project was born.

which can be done directly at the level of the action. After we do so, we are left with two transverse degrees of freedom, $X^2$ and $X^3$, and a longitudinal one, $\phi$. Generalizing our analysis of the 3D solid, we want to consider the ground-state solution

$$X^{2,3} = 0, \qquad \phi = \alpha\, x, \tag{3.4}$$

and small oscillations about it. As before, $\alpha$ measures the compression/dilation level of the bar and, if we leave at least one end free, the bar will relax to a particular value of $\alpha$, corresponding to zero tension or pressure.

To see this, we can look for instance at the $\phi$ equation of motion:

$$\delta(\phi - \phi^\star)\,\sqrt{g}\big[G(B) - 2BG'(B)\big] - 2\theta(\phi - \phi^\star)\,\partial_\alpha\big[\sqrt{g}G'(B)g^{\alpha\beta}\partial_\beta\phi\big] = 0\,. \tag{3.5}$$

Any configuration of the form (3.4) obeys the bulk part of this equation of motion, but the boundary part requires

$$G(B) - 2BG'(B) = 0\,, \qquad B = \alpha^2\,, \tag{3.6}$$

which is the one-dimensional analog of (2.13). Notice that this does correspond to a zero tension/pressure condition, since the world-sheet stress-energy tensor is

$$\tilde{T}_{\alpha\beta} = -2G'(B)\,\partial_\alpha\phi\partial_\beta\phi + g_{\alpha\beta}\,G(B)\,, \tag{3.7}$$

and so, on a configuration like (3.4), we have that the tension is

$$\mathcal{T} = -\tilde{T}_{11} = 2BG'(B) - G(B)\,, \tag{3.8}$$

which vanishes when eq. (3.6) is obeyed.

The vanishing of the tension in the ground state now directly implies the vanishing of a relevant Wilson coefficient in the excitations' action. This can be seen by expanding the action (3.2) around the ground state and using the tension relaxation condition (3.6); but, in fact, there is a deep structural reason behind it, which makes it manifest: the action (3.2) depends on the transverse embedding fields $\vec{\varphi} = (X^2, X^3)$ only through the induced metric $g_{\alpha\beta} \supset \partial_\alpha\vec{\varphi} \cdot \partial_\beta\vec{\varphi}$, which is quadratic in them. Thus, to quadratic order in these fields, it is enough to consider the expansion of the bar action to first order in induced-metric perturbations, which, *by definition*, yields the world-sheet stress energy tensor:

$$S_{\text{bar}} \supset \frac{1}{2}\int d^4x\,\tilde{T}^{\alpha\beta}\,\partial_\alpha\vec{\varphi} \cdot \partial_\beta\vec{\varphi}\,. \tag{3.9}$$

This directly implies that whenever the tension vanishes, so does the coefficient of $\vec{\varphi}'^2$, thus explaining the apparent fine tuning we described at the beginning of this section. This connection *is* enforced by symmetry, because it follows from the symmetry structure of the action. What is not a consequence of symmetry is the vanishing of the tension in the first place, which follows from leaving the boundary conditions free.

If one goes beyond the lowest-derivative action (3.2) and introduces higher-derivative terms, involving, for instance, the world-sheet extrinsic and intrinsic curvatures weighed by appropriate powers of the bar's thickness, one finds all possible higher-derivative corrections to the excitations' action, such as the $\vec{\varphi}''^2$ term in (3.1). However, none of this will affect eq. (3.9) and the associated connection between vanishing tension and vanishing coefficient for $\vec{\varphi}'^2$, since eq. (3.9) follows directly from the definitions of the induced metric and of the world-sheet stress-energy tensor.

## 4 Surface Tension Effects

Let us go back to the 3D case. The alert reader will have no doubt recognized that our analysis has not been systematically consistent, as the existence of the bounding $\theta$-function breaks the shift invariance acting on $\phi^I$. While bulk shift symmetry is preserved, there is nothing that stops us from writing down terms that are proportional to delta functions, and derivatives thereof, localized at the boundary of our solid. Such terms will be responsible for a surface tension, and we must consider their effect on our analysis and conclusions.

By appealing to some somewhat implicit geometric principle or intuition, surface tension in solids and liquids is usually modeled via a higher-dimensional, non-relativistic version of the Nambu-Goto action (see e.g. [17]). However, we believe that the most general symmetry-based characterization of surface tension and other surface terms will be more complicated than that. For instance, the example we studied in the previous section shows explicitly that, in the case of material submanifolds, there is more to physics than just the geometry of the submanifold: the field $\phi$, which has nothing to do with the embedding coordinates and thus with the geometry of the bar, is the one responsible for letting the tension of a bar relax to zero; in contrast, for the Nambu-Goto action the tension is a constant parameter.

We leave developing a general framework and understanding the systematics of this to future work. For the time being, we just want to get a sense of the order of magnitude we can generically expect for surface tension-like effects. To this end, let's trade the theta function cut-off for a smooth function $\theta_\ell$ with a finite thickness (membrane depth) $\ell$, say of order of the inter-atomic spacing, i.e. the UV cut-off. For instance, for a spherical solid of comoving radius $R$ we could choose

$$\theta_\ell\big(F(\phi)\big) = \frac{1}{2}\Big(1 + \tanh\frac{R - |\phi^I|}{\ell}\Big) = \frac{1}{1 + e^{-2(R-|\phi^I|)/\ell}} \,. \tag{4.1}$$

Now, the derivative expansion can be thought of as an expansion in the UV cutoff, $\ell$ in our case. Unfortunately, the function above is non-analytic in $\ell$ at $\ell = 0$. In fact, any $\ell$-dependent smooth function of $\phi$ that reduces to a $\theta$-function for $\ell \to 0$ must be non-analytic in $\ell$, since the $\theta$-function is trivial everywhere apart from its "jump"—$|\phi^I| = R$ in the example above—where it is singular. We thus have to interpret the $\ell \to 0$ limit and the associated expansion in powers of $\ell$ in a distributional sense.

In general, for a one-dimensional step-function of thickness $\ell$, we can expect an expansion of the form

$$\theta_\ell(x) = \theta(x) + C_1\,\ell\,\delta(x) + \frac{1}{2!}C_2\,\ell^2\delta'(x) + \dots \,, \tag{4.2}$$

where, barring accidents or symmetry reasons, the $C_n$ coefficients are dimensionless numbers of order one, as shown in Appendix C. It so happens that, owing to the tanh's being odd about its midpoint, for the explicit $\theta_\ell$ function above all $C_n$'s with odd $n$ vanish, but in more general cases they won't.

Let's then see what this implies in our case. For simplicity, let's stop at first order in $\ell$. We expect, for example,

$$\theta_\ell\big(F(\phi)\big) = \theta\big(F(\phi)\big) + C_1\,\ell\,\delta\big(F(\phi)\big)\,|\partial F| + \dots \,, \qquad C_1 = \mathcal{O}(1)\,, \tag{4.3}$$

where the extra factor of $|\partial F| \equiv \sqrt{\partial_\mu F \, \partial^\mu F}$ ensures that the result is independent of the choice of function $F$ we choose to parameterize a fixed surface. Notice that our choice for the norm of $\partial F$ is equivalent to using $B^{IJ} = \partial_\mu \phi^I \partial^\mu \phi^J$ as the inverse metric of comoving space. Consistently with the symmetries, we could have used, instead, $\delta^{IJ}$, or a linear combination of $\delta^{IJ}$ and $B^{IJ}$. These are inequivalent choices, which correspond to different regularizations of the $\theta$ function.

So, regularizing the $\theta$-function in (2.6) by giving it a finite thickness $\ell$ and allowing for a more general dependence on the fields close to the boundary, is equivalent to adding the boundary action

$$S_{\text{bdy}} = \ell \int d^4 x \, \delta(F(\phi)) \, |\partial F| \, \mathcal{L}_{\text{bdy}}\big(B^{IJ}, \phi^I\big) \,, \tag{4.4}$$

where we reabsorbed the $C_1$ coefficient of eq. (4.3) into $\ell$ and for now the boundary Lagrangian density $\mathcal{L}_{\text{bdy}}$ is undetermined. We emphasize once again that we do not know yet how the symmetries restrict the functional form of $\mathcal{L}_{\text{bdy}}$. The important aspects of this boundary action for us, however, are the explicit $\ell$ in front of it and the fact that, on physical and dimensional grounds, we can expect $\mathcal{L}_{\text{bdy}}$ to be of the same order of magnitude as the bulk Lagrangian density $G(B^{IJ})$ in (2.6), which, on a static configuration, is directly related to the energy density $\rho = T^{00}$ (see eq. (2.12)):

$$\mathcal{L}_{\text{bdy}} \sim G = -\rho \,. \tag{4.5}$$

In fact, for non-relativistic solids, apart from the rest-mass contribution to $\rho$, all entries of the stress-energy tensor are expected to be at most of order $\rho c_s^2$, where $c_s$ is the speed of sound, which can be much smaller than one. This implies that the above estimate for $\mathcal{L}_{\text{bdy}}$ is, in general, an upper bound.

Then, the static conservation equation (2.8) will be corrected by new boundary contributions, implying a relaxation condition like (2.10) but now with a nonzero r.h.s. However, this will be of order $\ell \rho$ times the dimensionally needed power of the typical length scale $R$ associated with the boundary of our solid, such as its radius of curvature. We thus expect the ground state to have a net tension which scales like

$$\tilde{T}_{ij} \sim \frac{\ell}{R} \rho \,, \tag{4.6}$$

which is hierarchically smaller than $\rho$ as long as our solid is much bigger than the UV cutoff $\ell$.

For example, if we take as a crude (but standard) model for the boundary terms (4.4) a constant energy density,

$$\mathcal{L}_{\text{bdy}}\big(B^{IJ}, \phi^I\big) = \text{const} \equiv -\Lambda \,, \tag{4.7}$$

corresponding to a constant surface tension $\sigma = \Lambda \ell$, we have a total stress-energy tensor of the form

$$T_{\mu\nu} = \theta\big(F(\phi)\big)\tilde{T}_{\mu\nu} - \ell \Lambda \, \delta(F(\phi))|\partial F| \, h_{\mu\nu} \,, \tag{4.8}$$

where, as before, $\tilde{T}_{\mu\nu}$ is the bulk stress-energy tensor, and $h_{\mu\nu}$ is the induced metric on the boundary in physical spacetime:

$$h_{\mu\nu} = \eta_{\mu\nu} - n_\mu n_\nu \,, \qquad n_\mu \equiv -\partial_\mu F/|\partial F| \,. \tag{4.9}$$

The static conservation equation (2.8) then gets modified to

$$\partial_i T^{i\nu} = \delta\big(F(\phi)\big)|\partial F| \left[ -n_i \, \tilde{T}^{i\nu} - \ell\Lambda \, \frac{1}{|\partial F|} \partial_i\big(|\partial F| h^{i\nu}\big) \right] + \theta\big(F(\phi)\big) \, \partial_i \tilde{T}^{i\nu} = 0 \,. \tag{4.10}$$

Notice that there is no $\delta'(F)$ term, thanks to the vanishing of $h^{\mu\nu}\,\partial_\mu F \propto h^{\mu\nu}\,n_\mu$. After some straightforward manipulations, and restricting for definiteness to the $\nu = j$ component, we find that the delta function term implies that at any point on the boundary we must have

$$n_i\,\tilde{T}^{ij} = \ell\Lambda\,n^j K \;, \tag{4.11}$$

where $K \equiv K^i{}_i$ is the trace of the boundary's extrinsic curvature tensor, $K^i{}_j = h^{ik}\partial_k n_j$. This is in perfect agreement with the estimate (4.6).

Alternatively, one can have reached the same approximate conclusion by a rough order of magnitude estimate: the contribution of a constant surface tension to the total action scales like the area of the boundary of our solid. In contrast, the bulk action scales like the volume. If the equations of motion imply a balance between boundary and bulk contributions, the relative importance of the surface tension must scale like the surface-to-volume ratio, that is, the typical extrinsic curvature of the boundary.

With hindsight, all this is obvious, at least empirically: whenever we quote the bulk properties of different materials—their density, thermal and electrical conductivities, etc.—to very good accuracy we do not expect these to depend on the overall size of the sample, or on its shape, as long as we are dealing with macroscopic objects[6]. The estimates above address this question quantitatively, at least as far internal stresses are concerned.

So, in conclusion, the surface tension balances the bulk stress, which will scale like the cut-off over the radius of curvature. One could protest that we have inserted a small number $\ell/R$, in violation of naturalness, in our action. Indeed, where does this scale $R$ come from? However, if choosing $R \gg \ell$ is a fine tuning in this field theory, then all macroscopic objects are finely-tuned! If we take that attitude, it would seem that the standard model hierarchy problem, or even the cosmological constant problem is just the tip of the (finely tuned) iceberg when it comes to fine tunings in nature.

## 5 The superfluid and the supersolid

So far we have shown that bounded target space field theories can lead to apparent fine tunings. Can we generalize this mechanism to relax, say, the cosmological constant? To do so, we will need to consider the time-like version of what we discussed above, that is, the combined spontaneous breaking of time translations and of a $U(1)$ symmetry down to their diagonal combination. This can be accomplished by considering a superfluid. Its symmetries are non-linearly realized by a single Goldstone field $\phi$, with an action of the form [18, 19]

$$S = \int d^4x\, P(X) \;, \qquad X \equiv -\partial_\mu\phi\,\partial^\mu\phi \;, \tag{5.1}$$

where $P$ is a generic function determined by the equation state. In particular, the ground state at chemical potential $\mu$ corresponds to a configuration of the form

$$\phi(x) = \mu t \;, \tag{5.2}$$

---

[6]Pushing this line of reasoning to its logical conclusions brings us dangerously close to the idea that all different shapes of pasta must taste the same.

and $P(\mu^2)$ is the associated pressure.

We can thus think of a superfluid as a "time solid," with the chemical potential $\mu$ in (5.2) playing the role of the scale factor $\alpha$ in (2.5). The stress energy tensor is given by

$$\tilde{T}_{\mu\nu} = 2P'(X)\partial_\mu\phi\partial_\nu\phi + \eta_{\mu\nu}P(X) \ . \tag{5.3}$$

In analogy with the case of the solid, we can now impose that the there is a boundary in $\phi$ space, for instance at some large $\phi = \phi^\star$. We thus have to multiply the Lagrangian density in (5.1) by

$$\theta(F(\phi)) = \theta(\phi^\star - \phi) \ . \tag{5.4}$$

Then, on the ground state solution (5.2), we find that

$$\tilde{T}^{0\nu}(\mu t = \phi^\star) = 0 \ , \tag{5.5}$$

implying, in particular,

$$\tilde{T}^{00}(x) = 2\mu^2 P'(\mu^2) - P(\mu^2) = 0 \ . \tag{5.6}$$

*everywhere.*

*If* the function $P(X)$ is such that this condition can be obeyed for some positive $X = \mu^2$, then the vacuum energy automatically vanishes, as a consequence of the boundedness of the target space. Notice that this mechanism is independent of the choice for $\phi^\star$. If we include a boundary action as well, then, as in the case of the solid, the ground state energy can in principle become non-zero. However, for a *flat* boundary at $t =$ const., this effect will be absent given that the boundary's extrinsic curvature vanishes in that case.

The interpretation of our result (5.6) is not as clear as in the case with a spatial boundary. There, if the boundary conditions are not met at some initial time, the solid will undergo oscillations and, assuming it can dissipate energy, will asymptotically reach the state that does obey the boundary conditions. But the superfluid can not be understood in this way. It is obvious that what we have found should not be called a "relaxation mechanism," since the boundary condition only applies at one point in time. To get a handle on this let us consider imposing initial conditions rather that final ones by putting the $\phi^\star$ boundary in the past. So, our action will be

$$S = \int d^4x \, P(X) f(\phi - \phi^\star) \ . \tag{5.7}$$

The only $\vec{x}$-independent solution seems to be the one with $T_{00} = 0$, at all times. Which is to say, there is no relaxation: the vacuum energy starts at zero and remains so.

Finally, we can combine the solid and the superfluid into a supersolid [20, 21]. This is a solid with an additional broken $U(1)$ symmetry at finite chemical potential, so that its dynamics can be described in terms of the solid comoving coordinates $\phi^I$ and the superfluid Goldstone $\phi$. The symmetries act on these fields in precisely the same ways as they act separately in the solid case and in the superfluid one. By bounding all fields through some general $\theta$ function that selects a hypersurface in $(\phi, \phi^I)$ space,

$$\theta\big(F(\phi, \phi^I)\big) \ , \tag{5.8}$$

and going through the same steps as above, we find that, on physically homogeneous configurations of the form

$$\phi = \mu t \ , \qquad \phi^I = \alpha x^I \ , \tag{5.9}$$

*all* components of the stress energy tensor must vanish. Is this the seed for a solution to the cosmological constant problem?

## 6 Conclusions

It has been a generally accepted tenet of quantum field theory that parameters in the action that are not protected by symmetries should take on values of order of the cut-off of the theory raised to the appropriate power. As is well appreciated, this situation can be avoided if the parameter is taken to be dynamical and allowed to relax. It would seem that, in general, generating such a mechanism takes insightful model building. Here however, we have shown that it can be quite generic, if one allows oneself to break space-time symmetries, and bound the target space. The former assumption is not a speculative stretch in the sense that our universe certainly has this property, but the latter assumption is less generic.

It is reasonable to question the technical naturalness of some of our assumptions—for example, the fact that we require very large volumes in field space. Still, the fact that we are literally surrounded by solid objects with precisely this property, many of which having a very natural origin, suggests that most of our assumptions are warranted, and natural in any meaningful sense of the word. Perhaps, when we move to the superfluid/supersolid case, which requires a boundary in the *time*-like field direction, some of these assumptions are on less solid footing. Clearly, the superfluid/supersolid case deserves further study, especially because of its potential relevance for the CC problem.

We close with a few important points:

1. Throughout the paper we have been dealing only with classical equations of motion and classical solutions, which, at first sight, seems to suggest that we have not addressed at all questions of radiative stability of our results.

   However, the transition to the quantum case is trivial: we just have to replace our classical actions ($S$) with the corresponding quantum effective actions ($\Gamma$), our fields with their expectation values, and our stress-energy tensors with their expectation values. Since to lowest order in derivatives the quantum effective actions have, as functions of the fields' expectation values, the same *local* structure as the classical actions they come from, our results are valid to all orders in perturbation theory, and, in fact, non-perturbatively.

   The reason this works is that all the field theories we have considered enjoy shift symmetries (up to boundary effects) for all their fields, and in that case the derivative expansion can be organized so that $\partial\phi$ is of order one, while higher derivatives are suppressed. So, the role usually played by the effective potential—the quantum effective action to zeroth order in the derivative expansion—is now played by the quantum effective action truncated to the level of one derivative per field, which is all one needs in order to study states that feature constant expectation values for the first derivatives of the fields.

This is reviewed in more detail in ref. [22], where an explicit one-loop analysis is also carried out [7].

2. We have made no claims regarding the CC problem as we have not coupled the system to gravity. Nor have we considered the vacuum effects of other fields. We could imagine using our Goldstone mode as a "control field", which couples to all the terms in the action, such that the full effective action includes the constraint on the field value. In this case we would still get a vanishing expectation value for the stress tensor.

We could also consider the case where we include a cosmological constant term to the action which does not include any couplings to the control field. This would lead to a modified version of our relaxation condition

$$T_{\text{in}}^{ij} = T_{\text{out}}^{ij} = -\Lambda_{\text{cc}}\delta^{ij} , \qquad T_{\text{in}}^{00} = T_{\text{out}}^{00} = \Lambda_{\text{cc}} , \tag{6.1}$$

where 'in' and 'out' denote whether we are inside or outside our media. This makes using our mechanism to solve the CC problem more challenging: first one has to solve the CC problem outside the medium, and then one has to find a viable version of our relaxation mechanism. However, assuming we live on a cosmologically large supersolid island, the physics outside the island can in principle be so different from the standard model's (unbroken SUSY?), that it is not inconceivable that it lead to a naturally small cosmological constant.

3. On a more prosaic note, we have shown, in a very physical context, that the bounding of the field values can lead to very novel renormalization group behavior, as we have seen in the dimensional reduction of a solid. RG flow due to KK reduction is typically trivial at weak coupling, in that the behavior of the two point function follows simple dimensional analysis. However, in the context of the class of theories we have considered here, KK reduction leads to a significant change in the two point function, even in the free field theory limit! In particular as we scale into the IR a higher derivative irrelevant operator becomes marginal. This behavior is all due to the bounded nature of the field, which introduces a scale into the problem in a novel fashion.

4. The relaxation mechanism will work with *any* current in the theory. For instance, if we had a finite (in space) solid with an internal symmetry group $G$, then the ground state would necessarily have a vanishing current density. Of course, this would only be of interest in systems which break rotational symmetry. There are clearly many other avenues in this direction that might be useful to explore.

5. Here we have studied two aspects of the field theory mechanism we have considered. On the one hand, it gives a way to set expectation values of currents to zero. This may or may not be of interest depending upon the context. In the case of time solids we were motivated by opening, perhaps, a new window on the CC problem. On the other hand, in the case of

---

[7]In our case, to calculate the quantum effective action one certainly would not want to calculate with the action that vanishes for field values outside the boundary. Instead, one would calculate in the Eulerian picture, where one works instead with the $X^I(\phi^I, t)$ fields, in which case the boundary effects are easily calculated. One can then map back to the Lagrangian picture to get the effective action as a functional of $\phi^I$ (see e.g. [7]).

spatial solids, we were motivated by explaining fine tunings in an action. Even in the case of time solids the constraints on the vacuum energy will lead to relations between couplings in the action that are not protected by symmetry.

## Acknowledgments

We would like to thank A. Acharya, C. Cheung, G. Cuomo, B. Freivogel, F. Piazza for useful discussions and comments. AN is partially supported by the US DOE (award number DE-SC011941) and by the Simons Foundation (award number 658906). IZR is supported by DOE grants DE-FG02-04ER41338 and FG02-06ER41449.

## A    Constraints on the dynamics of a relaxed 3D solid

Consider the general expansion (2.16) for the bulk action of a 3D solid around the state (2.5). Using the relaxation condition (2.13) and dropping a constant term in (2.16), we are left with

$$S_{\text{bulk}} = \int d^4x \left( \frac{1}{2\alpha^2} G_0 s^{II} + \frac{1}{2} \frac{\partial G}{\partial B^{IJ} \partial B^{KL}} \bigg|_0 s^{IJ} s^{KL} + \dots \right) , \tag{A.1}$$

where the second term and all the subsequent ones are constrained by the symmetries of the action and of the ground state, but *not* by the relaxation condition. The structure of the first term, on the other hand, is completely determined by the relaxation condition (2.12), which has been utilized in this equation. Still, since the expansion of $s^{IJ}$ stops at quadratic order in the fluctuation field $\pi^I$—see (2.17)—such a term only affects the action up to second order, and so the consequences of the relaxation condition stop at quadratic order, at least as far the bulk dynamics are concerned. What are they?

To answer this question, we expand all the terms above up to second order in $\pi^I$, and ignore a total derivative linear term ($\propto \vec{\nabla} \cdot \vec{\pi}$). We get:

$$S_{\text{bulk}} \simeq \int d^4x \left[ \frac{1}{2} G_0 \, \partial_\mu \pi^i \partial^\mu \pi^i + 2C_0 \, (\vec{\nabla} \cdot \vec{\pi})^2 + 2D_0 \, \partial_{(i} \pi_{j)} \partial_{(i} \pi_{j)} \right] , \tag{A.2}$$

where we have parametrized the second derivatives of $G(B^{IJ})$ in the most general way compatible with isotropy,

$$\frac{\partial G}{\partial B^{IJ} \partial B^{KL}} \bigg|_0 = \frac{1}{\alpha^4} \big( C_0 \, \delta_{IJ} \delta_{KL} + D_0 \, \delta_{I(K} \delta_{L)J} \big), \tag{A.3}$$

with generic $C_0$ and $D_0$, and we stopped differentiating between internal ($I$-type) and spatial ($i$-type) indices, because the unbroken rotations act on them in the same way.

Upon integrating by parts, one recognizes that this action is nothing but the most general quadratic action consistent with the symmetries:

$$S_{\text{bulk}} \simeq -\frac{1}{2} G_0 \int d^4x \left[ \dot{\vec{\pi}}^2 - (c_L^2 - c_T^2)(\vec{\nabla} \cdot \vec{\pi})^2 - c_T^2 \, \partial_i \pi_j \partial_i \pi_j \right] , \tag{A.4}$$

where $c_L$ and $c_T$ are the longitudinal and transverse phonon speeds,

$$c_L^2 = 1 + \frac{4(C_0 + D_0)}{G_0} , \qquad c_T^2 = 1 + \frac{2D_0}{G_0} . \tag{A.5}$$

To be more precise, there is one constraint on this action that does not follow purely from the symmetries. It is that the overall coefficient that weighs the kinetic energy must be $-G_0$, the ground state's rest energy density: an interesting statement, but perhaps not too surprising.

Notice that getting to the above conclusion involved integrating by parts and ignoring boundary terms. So, it is instructive to think about all this in the presence of our boundary. To this end, before performing any integration by parts, we also expand the $\theta$-function in (2.6) in powers of $\pi^I$. Stopping at quadratic order, and taking into account the $\theta$-function and its derivatives, we can integrate by parts at will and ignore the boundary terms at infinity—those will certainly vanish. If we move all derivatives away from the $\theta$-function, we end up with the quadratic action

$$S \simeq -\frac{1}{2}G_0 \int d^4x\, \theta\big(F(\alpha\vec{x})\big) \left[\dot{\vec{\pi}}^2 - (4C_0/G_0 - 1)(\vec{\nabla}\cdot\vec{\pi})^2 - (2 + 4D_0/G_0)\partial_{(i}\pi_{j)}\partial_{(i}\pi_{j)}\right] \qquad \text{(A.6)}$$

$$= -\frac{1}{2}G_0 \int d^4x\, \theta\big(F(\alpha\vec{x})\big) \left[\dot{\vec{\pi}}^2 - (c_L^2 - 2c_T^2)(\vec{\nabla}\cdot\vec{\pi})^2 - 2c_T^2\,\partial_{(i}\pi_{j)}\partial_{(i}\pi_{j)}\right]\,, \qquad \text{(A.7)}$$

where the various parameters are defined exactly as above.

Now it looks like that there is a nontrivial constraint: the action, written with an undifferentiated $\theta$, does not depend on the *antisymmetric* spatial derivatives of $\pi^i$; in contrast, a term like $\left(\partial_{[i}\pi_{j]}\right)^2$ would have been perfectly consistent with the symmetries.

As our bulk analysis above shows, however, this constraint is invisible in the bulk. This means that, upon integrating by parts, we can rewrite the action as the most general bulk one (A.4) plus a boundary term, and it is this boundary term that happens to be constrained by the relaxation condition. Indeed, we find:

$$S \simeq -\frac{1}{2}G_0 \int d^4x\, \left\{\theta(F)\left[\dot{\vec{\pi}}^2 - (c_L^2 - c_T^2)(\vec{\nabla}\cdot\vec{\pi})^2 - c_T^2\,\partial_i\pi_j\partial_i\pi_j\right]\right. \qquad \text{(A.8)}$$

$$\left. + \delta(F)\vec{\nabla}F\, c_T^2\left[(\vec{\pi}\cdot\vec{\nabla})\vec{\pi} - \vec{\pi}(\vec{\nabla}\cdot\vec{\pi})\right]\right\}\,. \qquad \text{(A.9)}$$

The boundary term can be written in more geometric terms:

$$S_{\text{bdy}} = \frac{1}{2}G_0 c_T^2 \int dt \oint d\Sigma\, \hat{n}\cdot\left[\vec{\pi}\times(\vec{\nabla}\times\vec{\pi})\right] = \frac{1}{2}G_0 c_T^2 \int dt \oint \pi\wedge\star d\pi, \qquad \text{(A.10)}$$

where $d\Sigma$ is the area element on the boundary, and $\hat{n}$ the outgoing normal.

We can thus think of the constraint coming from the relaxation condition as the following statement: for a relaxed solid, if we decide to write the bulk action in the standard form (A.8), there must be an accompanying boundary term of the form (A.10). This boundary term ensures that the action is invariant under rotations of the ground state, $\langle\phi^I\rangle = x^I \to \langle R^I{}_J\phi^J\rangle \simeq x^I + \theta^a x^b \epsilon^{abI}$. That is, if we take $\pi^I = \theta^a x^b \epsilon^{abI}$ the action of the boundary term cancels the antisymmetric part of $\partial_i\pi_j\partial_i\pi_j$. Note however, that this cancellation is a manifestation of the non-linearly realized broken internal rotational symmetry and not the boundedness of the target space. If we stressed the sample, even anisotropically, we would still have rotational symmetry, but now the boundary action would change to reflect the stress. If we have an anisotropic lattice however, and we stress the sample in a non-isotropic fashion, the energy will change under rotations of the ground state. So the manifestation of the open boundary in three dimensions, is to simply pick out a particular operator on the boundary to ensure rotational symmetry. In the case of the one dimensional solid however, the open boundary conditions lead to a local (bulk) effect.

# B    The bar as an EFT puzzle: dimensional reduction

Consider a relaxed rectangular solid. As discussed above, it will feature a gradient energy $\sim (\partial\pi)^2$ for all of its perturbations $\vec{\pi}$. Now consider shrinking some of the sides to form a rod (or bar, or beam—we will use these characterizations interchangebly). As proved in sect. 3, in the dimensionally reduced theory the transverse oscillations $\vec{\varphi}$ do not have a standard gradient energy, but only a higher-derivative one that scales as $(\partial^2\vec{\varphi})^2$.

From the viewpoint of EFT, this is rather puzzling since typically dimensional reduction does not change the low-energy dispersion relation in this way: one might get a KK mass term in addition to the original gradient energy, which can be thought of as a relevant deformation of the original theory, but not a replacement of the original gradient energy with an apparently *irrelevant* higher-derivative one. In this sense, this theory has a very unusual renormalization group trajectory. The problem does not lie with the reduced theory but with the full theory: the reduced theory is only valid for length scales much longer than the thickness of the rod, but the full theory must be treated with care at long distances.

We can gain some intuition by understanding how the energetics change as we match from the full theory to the IR one. Physically, the vanishing of the standard gradient energy for a bar is a consequence of leaving open boundary conditions and an asymmetry in the geometry, as discussed in [9], for example . However, given our setup and results, we can make this more clear by explicitly performing the dimensional reduction. For simplicity, we consider a rectangular solid in 2+1 dimensions. We will call $L$ the long side (along $x$) and $\delta$ the short one (along $y$). We are interested in the case $L \gg \delta$. We can even take the infinite $L$ limit, as long as we let the solid relax to zero tension. The quadratic action thus is eq. (A.7), with $\theta(F)$ nonzero only within our $L$-by-$\delta$ rectangle, which we call $\mathcal{R}$.

In fact, it is instructive to keep the relative coefficients of the quadratic action generic,

$$S = \frac{1}{2} \int_{\mathcal{R}} dxdydt \left( \dot{\vec{\pi}}^{\,2} - a\big(\vec{\nabla}\cdot\vec{\pi}\big)^2 - b(\partial_i\pi_j)(\partial_j\pi_i) - c(\partial_i\pi_j)(\partial_i\pi_j) \right) , \tag{B.1}$$

and refrain from using their actual values,

$$a = c_L^2 - 2c_T^2 , \qquad b = c = c_T^2 , \tag{B.2}$$

until the end of the computation. Notice that, given that $c_L$ and $c_T$ are generic, the only real constraint coming from the relaxation condition is $b = c$. Notice also that we are ignoring the overall normalization of the action, which is irrelevant for classical questions.

Now we come to the interesting twist compared to more standard dimensional reductions: the fact that we are leaving the boundary conditions open, implies that we cannot perform a mode expansion such as a KK decomposition and retain only the zero modes. In fact, the zero modes do not correspond to allowed solutions.

To see this, we can vary the action without imposing that the variations vanish at the boundary. The variation of the action then has a bulk term and a boundary term, and these must separately vanish. The boundary term is

$$-\oint_{\partial\mathcal{R}} \hat{n}_i\, \delta\pi_j \left[ a\delta_{ij}\, \vec{\nabla}\cdot\vec{\pi} + b\, \partial_j\pi_i + c\, \partial_i\pi_j \right] . \tag{B.3}$$

Imposing that this vanish along the long boundaries, $y = \pm\delta/2$, $\hat{n} = \pm\hat{y}$, we get the constraints

$$b\,\partial_1\pi_2 = -c\,\partial_2\pi_1 \qquad \text{at } y = \pm\frac{\delta}{2} \tag{B.4}$$

$$a\,(\vec{\nabla}\cdot\vec{\pi}) = -(b+c)\,\partial_2\pi_2 \qquad \text{at } y = \pm\frac{\delta}{2}\,. \tag{B.5}$$

There is a similar set of constraints coming from the short sides at $x = \pm L/2$. However, for $L \to \infty$ we can ignore them for local physics far from those sides.

Since these constraints at the boundary relate spatial derivatives along $y$ to spatial derivatives along $x$, we see that the usual KK decomposition cannot work. In particular, zero modes along $y$ cannot correspond to wave solutions with finite $k_x$.

We thus take a different approach: we solve the bulk equations of motion for given boundary conditions, plug the solutions back into the action, and obtain in this way an effective action for the boundary values of our fields. At low $x$-momenta, in a derivative expansion, this boundary effective action will be local. Essentially, we are writing the partition function as

$$Z = \int_{\partial\mathcal{R}} D\pi_i \int_{\mathcal{R}} D\pi_i\, e^{iS} \tag{B.6}$$

and performing the innermost path-integral for given boundary values of the fields.

Integration by parts leads to an action $S = S_{\text{bulk}} + S_{\text{bdy}}$, with

$$S_{\text{bulk}} = \frac{1}{2}\int_{\mathcal{R}} dx\,dy\,dt\,\vec{\pi}\cdot\left(-\ddot{\vec{\pi}} + (a+b)\,\vec{\nabla}(\vec{\nabla}\cdot\vec{\pi}) + c\,\nabla^2\vec{\pi}\right)$$

$$S_{\text{bdy}} = \frac{1}{2}\int dx\,dt\left[-a\,\pi_2(\vec{\nabla}\cdot\vec{\pi}) - b\,\vec{\pi}\cdot\vec{\nabla}\pi_2 - c\,\pi_j\partial_2\pi_j\right]_{y=-\delta/2}^{y=\delta/2},$$

where we are taking the $L \to \infty$ limit and thus neglect contributions from those faraway boundaries. Now the idea is simply to use the bulk equations of motion as well as the boundary constraints we derived above to derive an effective theory that depends only on the boundary fields and their $x$- and $t$-derivatives.

We start by noticing that the bulk part of action, written as in (B.7), vanishes on the equations of motion (this is a general property of quadratic actions), and so we are just left with the boundary action (B.7). This is the difference of a certain $y$-dependent quantity $Q(y)$ evaluated on the top side ($y = \delta/2$) and the same quantity evaluated at the bottom side ($y = -\delta/2$). The analog of focusing on the zero mode, now, seems to be to perform a Taylor expansion in $y$ of such a quantity: $Q(y) = Q(0) + yQ'(0) + \dots$. For a wave of $x$-momentum $k$, all spatial derivatives will be or order $k$, and so such an approximation should be correct at lowest order in $k\delta$.

We thus get the dimensionally reduced effective action

$$S_{\text{eff}} \simeq \frac{\delta}{2}\int dx\,dt\,\partial_2\left(-a\,\pi_2(\vec{\nabla}\cdot\vec{\pi}) - b\,\vec{\pi}\cdot\vec{\nabla}\pi_2 - c\,\vec{\pi}\cdot\partial_2\vec{\pi}\right) \tag{B.7}$$

We now want to eliminate derivatives along $y$ in favor of the other derivatives. To this end, we write the bulk equations of motion as

$$\partial_2^2\pi_1 = \frac{1}{c}[\ddot{\pi}_1 - (a+b+c)\partial_1^2\pi_1 - (a+b)\,\partial_1\partial_2\pi_2]$$

$$\partial_2^2\pi_2 = \frac{1}{a+b+c}[\ddot{\pi}_2 - c\,\partial_1^2\pi_1 - (a+b)\,\partial_1\partial_2\pi_1]$$

and the boundary constraints as

$$\partial_2 \pi_1 = -\frac{c}{b} \partial_1 \pi_2 \tag{B.8}$$

$$\partial_2 \pi_2 = -\frac{a}{a+b+c} \partial_1 \pi_1 . \tag{B.9}$$

Putting everything together, and freely integrating by parts along $x$ and $t$, we finally find the effective action for our $x$- and $t$-dependent fields:

$$S_{\text{eff}} \simeq \frac{\delta}{2} \int dx dt \left[ \dot{\vec{\pi}}_i^2 - \left( \frac{b^2}{c} - c \right)(\partial_1 \pi_2)^2 - (\partial_1 \pi_1)^2 \left( a + b + c - \frac{a^2}{a+b+c} \right) \right] . \tag{B.10}$$

Now we can use the specific values (B.2) for $a$, $b$, and $c$. In particular $b$ and $c$ are equal, and this makes the propagation speed of the transverse mode $\pi_2$ vanish at this order in derivatives! This is of course what we were expecting, given the analysis of sect. 3, but it provides a quite nontrivial cross-check of our formalism and results.

Also interesting is the fact that the longitudinal mode, $\pi_1$, has a lower propagation speed than it had in the bulk. We can summarize the RG flow of the speeds as:

$$c_T^2 \to 0 , \qquad c_L^2 \to \bar{c}_L^2 \equiv 4c_T^2 (1 - c_T^2/c_L^2) . \tag{B.11}$$

As an independent check, we computed the phonons' two-point function on the infinitely long strip of thickness $\delta$. For simplicity, we restricted to $t$-indepedent and $y$-dependent sources, so the two-point function we are talking about actually is

$$G^{ij}(x) \equiv \int dt dy dy' \langle T\pi^i(x, y, t)\pi^j(0, y', 0) \rangle , \tag{B.12}$$

where the $y$ and $y'$ integrals extend between $-\delta/2$ and $+\delta/2$. After a tedious Green's function computation, aided by Mathematica, we find

$$\tilde{G}^{12} = \tilde{G}^{21} = 0 \tag{B.13}$$

$$\tilde{G}^{11}(k) = -i\frac{\delta}{c_L^2 k^2} \left[ 1 + \frac{(c_L^2 - 2c_T^2)^2}{(c_L^2 - c_T^2)c_T^2} f_+(k\delta) \right] \tag{B.14}$$

$$\tilde{G}^{22}(k) = -i\frac{\delta}{c_T^2 k^2} \left[ 1 + \frac{c_L^2}{c_L^2 - c_T^2} f_-(k\delta) \right] , \tag{B.15}$$

where

$$f_\pm(\xi) \equiv \frac{2\sinh^2(\xi/2)}{(\sinh\xi \pm \xi)\,\xi} . \tag{B.16}$$

In the deep UV, we recover the correct bulk behavior of the static two-point functions:

$$\tilde{G}^{11}(k \gg 1/\delta) \simeq -i\frac{\delta}{c_L^2 k^2} , \qquad \tilde{G}^{22}(k \gg 1/\delta) \simeq -i\frac{\delta}{c_T^2 k^2} . \tag{B.17}$$

In the deep IR however, things change dramatically, especially for $\pi_2$:

$$\tilde{G}^{11}(k \ll 1/\delta) \simeq -i\frac{\delta}{\bar{c}_L^2 k^2} , \qquad \tilde{G}^{22}(k \ll 1/\delta) \simeq -i\frac{\delta}{\bar{c}_L^2/c_L^2} \frac{1}{\delta^2 k^4} , \tag{B.18}$$

where the infrared longitudinal speed $\bar{c}_L^2$ is the same as defined above. We thus see that in moving from the UV to the IR, the transverse propagator changes from $\sim 1/k^2$ to $\sim 1/k^4$, in agreement with all that we have discussed above.

## C  A distributional expansion

Consider a generic regularization of the $\theta$ function, $\theta_\ell$, with small thickness $\ell$. For definiteness, we can take

$$\theta_\ell(x) = f(x/\ell) , \qquad (\ell > 0) , \qquad (C.1)$$

where $f$ is a regular function with order-one variations and derivatives, that rapidly goes to zero at $-\infty$ at to one at $+\infty$.

To understand the properties of $\theta_\ell$ as a distribution in the small $\ell$ limit, we consider its action on a generic smooth, rapidly decaying test function $\varphi(x)$. In particular, we are interested in how different $\theta_\ell$ is from the $\theta$ function itself. So:

$$\langle (\theta_\ell - \theta), \varphi \rangle \equiv \int dx \left( f(x/\ell) - \theta(x) \right) \varphi(x) \qquad (C.2)$$

$$= \int dy \left( f(y) - \theta(y) \right) \ell \, \varphi(y\ell) \qquad (y = x/\ell) \quad (C.3)$$

$$= \int dy \left( f(y) - \theta(y) \right) \left[ \ell \, \varphi(0) + \ell^2 y \, \varphi'(0) + \frac{1}{2!} \ell^3 y^2 \, \varphi''(0) + \dots \right] \qquad (C.4)$$

$$= C_1 \, \ell \, \langle \delta, \varphi \rangle + \frac{1}{2!} C_2 \, \ell^2 \, \langle \delta', \varphi \rangle + \frac{1}{3!} C_3 \, \ell^3 \, \langle \delta'', \varphi \rangle + \dots , \qquad (C.5)$$

where we have used $\theta(y\ell) = \theta(y)$, we have assumed that our test function $\varphi$ can be expanded in a Taylor series, and we have defined the $C_n$ coefficients as

$$C_n \equiv n \int dy \big( f(y) - \theta(y) \big) (-y)^{n-1} , \qquad (C.6)$$

which, given our assumptions on $f$, generically yield order-one numbers. We thus recover exactly the expansion in (4.2).

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
