# Peer review of "Apparent Fine Tunings for Field Theories with Broken Space-Time Symmetries"

_SciPost Physics_

## Round 1 · Referee Report · Anonymous · 2023-2-18

Strengths

1- original and creative contribution
2- possible wide spectrum of applicability (cond-mat, cosmology, etc)
3- well written
4- concrete examples

Weaknesses

1- immediate physical predictions not clear
2- relevance for real condensed matter systems not obvious

Report

This work considers zero temperature effective field theories for solids, supersolids and superfluids with open boundary conditions. As a result of these modified boundary conditions, the EFTs have hierarchically small Wilson coefficients even if they are not protected by symmetries nor finely tuned by hand. Most of the discussion is oriented towards condensed matter applications but possible, and fundamental, impact for cosmology is discussed as well. The manuscript is well-written, with concrete examples and contains interesting and original results. I believe that this work should be published after some revisions are considered.

Below, I list a number of comments/suggestions/questions.

(1) In the introduction, the authors write "Those that are protected by symmetry, but which take on smaller values than
expected from dimensional analysis,''. What do they mean by that? Dimensional analysis is not able to fix a value for a coefficient. Do they mean that one naturally expects O(1) values for dimensionless coefficients? Please clarify.

(2) If I understand correctly, following from the open boundary conditions, the stress relaxes to zero. Is this condition anything else than mechanical equilibrium for a system able to transmit stress with an environment? Expand about the physical meaning of the minimization condition 2.13.

(3) Is Ref. [8] in footnote 1 the correct reference? To my knowledge, the coset construction for solids is much older than that. For example, 1307.0517. Please fix it.

(4) Eqs. 2.8 and 2.9 are the conservation of the stress-tensor. If the system is an open system, able to interact with an external environment, why its stress tensor should be conserved? Please expand on this point.

(5) Is it obvious that the preferred phase with open boundary conditions should retain isotropy?

(6) Is it correct to say that now, because of open boundary conditions, the value of alpha in Eq.2.5 is dynamically determined (following the minimization of the potential in 2.14)? If so, what does alpha correspond to in say a solid? is it the lattice spacing scale? In the case of a superfluid, it is the chemical potential that obeys a similar relation, Eq. 5.6. I find weird that the chemical potential, which is usually thought as an external source for the corresponding charge density, is dynamically determined. Could the authors explain this better?

(7) It seems to me that the action in Eq.(3.1) has an emergent "fracton-like" polynomial shift symmetry, $\varphi \rightarrow \varphi +c+b_i x^i$. This emergent symmetry kills the appearance of first derivative terms in space and finely tunes the corresponding Wilson coefficient to zero. That is clearly not a symmetry of the UV theory, but it seems an IR emergent symmetry. That would also explain the IR dispersion $\sim k^4$ described in the appendix in Eq.(B.18) and the curious RG flow properties of the theory, similar to those discussed in the fractons business. Is there anything more than a naive analogy? Can the example of the oscillating bar be explained in this way? Also, and more generally, is this hierarchically small coefficient simply a consequence of an emergent symmetry in the IR which probably does not appear in the case of closed boundary conditions?

(8) In the appendix, the authors discuss the linear excitations around the relaxed equilibrium configuration. Is there any difference between them and those in a solid with closed boundary conditions? Can the dispersion relation of the low-energy phonons detect the different boundary conditions?

Requested changes

1- clarifications and minor revisions related to my points above are required

  • validity: top
  • significance: high
  • originality: high
  • clarity: high
  • formatting: excellent
  • grammar: perfect

Author:  Ira Rothstein  on 2023-03-10  [id 3465]

(in reply to Report 1 on 2023-02-18)

We thanks the referee for the constructive comments: Below please find our replies:

``(1) In the introduction, the authors write "Those that are protected by symmetry, but which take on smaller values than
expected from dimensional analysis,''. What do they mean by that? Dimensional analysis is not able to fix a value for a coefficient. Do they mean that one naturally expects O(1) values for dimensionless coefficients? Please clarify.''

Yes by dimensional analysis we just mean order one in units of the cut-off.

``(2) If I understand correctly, following from the open boundary conditions, the stress relaxes to zero. Is this condition anything else than mechanical equilibrium for a system able to transmit stress with an environment? Expand about the physical meaning of the minimization condition 2.13."

There is no transmission of stress. Its a closed system. The minimum of the effective action automatically has zero stress.

``(3) Is Ref. [8] in footnote 1 the correct reference? To my knowledge, the coset construction for solids is much older than that. For example, 1307.0517. Please fix it.''

Yes that should have been [7]. Thats been fixed, thanks for pointing that out.

``(4) Eqs. 2.8 and 2.9 are the conservation of the stress-tensor. If the system is an open system, able to interact with an external environment, why its stress tensor should be conserved? Please expand on this point.''

The system is not open.

``(5) Is it obvious that the preferred phase with open boundary conditions should retain isotropy?''

In generally the boundary on the field will break some of the symmetries. But his has no effect on the bulk physics only the surface physics, which is suppressed as explained in section 4.

``(6) Is it correct to say that now, because of open boundary conditions, the value of alpha in Eq.2.5 is dynamically determined (following the minimization of the potential in 2.14)? If so, what does alpha correspond to in say a solid? is it the lattice spacing scale? In the case of a superfluid, it is the chemical potential that obeys a similar relation, Eq. 5.6. I find weird that the chemical potential, which is usually thought as an external source for the corresponding charge density, is dynamically determined. Could the authors explain this better?''

Yes we agree with this statement. We also agree that it is weird in the case of the ``superfluid". The word superfluid here is only used colloquially. The superfluid analogy is only rough. As we state in the paper, this theory deserves further study.

``(7) It seems to me that the action in Eq.(3.1) has an emergent "fracton-like" polynomial shift symmetry,
φ→φ+c+bixi. This emergent symmetry kills the appearance of first derivative terms in space and finely tunes the corresponding Wilson coefficient to zero. That is clearly not a symmetry of the UV theory, but it seems an IR emergent symmetry. That would also explain the IR dispersion
k4 described in the appendix in Eq.(B.18) and the curious RG flow properties of the theory, similar to those discussed in the fractons business. Is there anything more than a naive analogy? Can the example of the oscillating bar be explained in this way? Also, and more generally, is this hierarchically small coefficient simply a consequence of an emergent symmetry in the IR which probably does not appear in the case of closed boundary conditions?''

That is only a symmetry of the dispersion relation, not the interactions. Otherwise indeed our result would not be interesting.

``(8) In the appendix, the authors discuss the linear excitations around the relaxed equilibrium configuration. Is there any difference between them and those in a solid with closed boundary conditions? Can the dispersion relation of the low-energy phonons detect the different boundary conditions?''

Yes, if the sample is stressed at the ends the dispersion will become canonical.

Anonymous on 2024-01-19  [id 4264]

(in reply to Ira Rothstein on 2023-03-10 [id 3465])

The Authors have satisfactorily replied to my previous concerns. I am happy with the current version of their manuscript.

---

## Round 1 · Referee Report · Anonymous · 2023-2-28

Report

This paper presents a mechanism which leads to the vanishing of certain coefficients in the low energy effective action for theories which break space-time symmetries. It is not a (direct) consequence of symmetry requirements, nor a fine-tuning; rather it arises from the relaxation of stresses or currents through open boundary conditions. This is a very interesting idea, with possibly very exciting applications to various fields of physics, and in particular to cosmology. Some simple yet illustrative examples are explicitly worked out in the manuscript.

However, I believe that some further dicsussion on the following points would be interesting:

(i) Very generally, in QFT, the effective action depends on which state one is expanding around of. So it does not sound surprising that, expanding around a state of zero stresses/currents, some Wilsonian coefficients vanish. I understand that the non-trivial point is the mechanism with which such a state is chosen: relaxation through open boundary conditions. This makes the choice natural (in a physical sense), instead of artificial. Do the authors agree with this statement? Does the paper go beyond this idea in any sense?

(ii) Similar observations hold in the context of finite-temperature hydrodynamic effective field theories. The transport coefficients are in general functions of the state variables, and expanding around zero or finite density states leads to different physics. Could the authors comment on similiraties/differencies between these settings? Relatedly, how would adding finite temperature affect the results?

(iii) Why are space-time symmetries crucial? There is a comment on internal symmetries in the conclusions, but wouldn't that be a much more natural and easy setup to illustrate the general idea?

(iv) The fact that the target space is bounded sounds crucial in the argument. However, Goldstone modes can also relax through pinning, i.e. by also breaking explicitly but perturbatively the relevant symmetry. Would such a bulk mechanism lead to the same results on the apparent fine-tuning in the effective actions?

(v) Finally, the superfluid example is quite curious (as also mentioned in the text). The superfluid phase is compact, so what does "boundary at some large $\phi=\phi^\star$" mean? Is there some natural separation of scales? Even if there is, the field $\phi$ is just the phase of the order parameter, so how can an "open boundary" in the phase of an operator by physically realized? Any clarifications on this point would be very useful.

In general, the paper is very well-written and contains interesting results, while the logical steps and the calculations are clear and easy to follow. So I would be very happy to suggest it for publication in SciPost provided that some further comments on the points above are added.

  • validity: top
  • significance: good
  • originality: good
  • clarity: top
  • formatting: perfect
  • grammar: perfect

Author:  Ira Rothstein  on 2023-03-10  [id 3464]

(in reply to Report 2 on 2023-02-28)

We thanks the referee for the constructive comments. Below please find our replies:

``(i) Very generally, in QFT, the effective action depends on which state one is expanding around of. So it does not sound surprising that, expanding around a state of zero stresses/currents, some Wilsonian coefficients vanish. I understand that the non-trivial point is the mechanism with which such a state is chosen: relaxation through open boundary conditions. This makes the choice natural (in a physical sense), instead of artificial. Do the authors agree with this statement? Does the paper go beyond this idea in any sense?''

We are not sure we agree with this statement in that bounding the target space does not correspond to a choice of states. This constraint forces the effective action to have a particular ground state which is the upshot of the relaxation mechanism. In other words, we are not “choosing a state to expand around”. The key point is that there is a relaxation mechanism, much as in the case of PQ.

``(ii) Similar observations hold in the context of finite-temperature hydrodynamic effective field theories. The transport coefficients are in general functions of the state variables, and expanding around zero or finite density states leads to different physics. Could the authors comment on similiraties/differencies between these settings? Relatedly, how would adding finite temperature affect the results?''

We dont believe that finite temperature effects are relevant, as the free energy will still relax to a state of zero tension. The action will acquire temperature dependent couplings but the same logic will follow leading to vanishing coefficients.

``(iii) Why are space-time symmetries crucial? There is a comment on internal symmetries in the conclusions, but wouldn't that be a much more natural and easy setup to illustrate the general idea?''

The breaking of space-time symmetries is crucial because we are considering a vev of derivatively coupled fields. We dont know how generate a relaxation mechanism, without breaking space-time symmetries.

``(iv) The fact that the target space is bounded sounds crucial in the argument. However, Goldstone modes can also relax through pinning, i.e. by also breaking explicitly but perturbatively the relevant symmetry. Would such a bulk mechanism lead to the same results on the apparent fine-tuning in the effective actions?''

In this case the symmetry associated with Goldstones, is translations (taking the GB’s to be phonons). Breaking the symmetry explicit will gap them, but will not lead to any vanishing Wilson coefficients.

``(v) Finally, the superfluid example is quite curious (as also mentioned in the text). The superfluid phase is compact, so what does "boundary at some large
ϕ=ϕ⋆
" mean? Is there some natural separation of scales? Even if there is, the field ϕ
is just the phase of the order parameter, so how can an "open boundary" in the phase of an operator by physically realized? Any clarifications on this point would be very useful.''

We are using the term superfluid here losely, as the field in this theory is not a phase.

Anonymous on 2023-03-13  [id 3470]

(in reply to Ira Rothstein on 2023-03-10 [id 3464])

I would like to thank the authors for their responses to my questions. I still have a few more comments and suggestions for improvement to make:

(i) "We are not sure we agree with this statement in that bounding the target space does not correspond to a choice of states. This constraint forces the effective action to have a particular ground state which is the upshot of the relaxation mechanism. In other words, we are not “choosing a state to expand around”. The key point is that there is a relaxation mechanism, much as in the case of PQ."

But I think that this is exactly what I said. My argument was that the "accidental fine tuning" is a sole consequence of the state one is expanding around. The relaxation mechanism is what naturally leads to the specific state, instead of it being chosen by hand. Do you agree with this? If yes, that should be more clear in the paper.

(ii) - (iv) Adding these comments in the paper would be useful.

(v) "We are using the term superfluid here losely, as the field in this theory is not a phase."

Since it is not clear what the physical system is, and also given that you only write down the relaxation condition but you do not derive the fine tunings, I am not sure what is the use of this section and how it belongs in the main body of the paper. It might be better to move this to an appendix, and maybe move appendix B in the main body, since it is quite interesting.

Anonymous on 2023-09-13  [id 3976]

(in reply to Anonymous Comment on 2023-03-13 [id 3470])

We thank the referee for the reply. We agree about the ground state and have added a comment on top of page 6. We have also added comments regarding ii-iv. We have moved Appendix B to the main body but ask that we keep the superfluid appendix in the main body since when giving talks this seems to generate a lot of interest.

Attachment:

Sbarrasubmit3.pdf

Anonymous on 2023-09-13  [id 3979]

(in reply to Anonymous Comment on 2023-09-13 [id 3976])

The previous file was missing the references.

Attachment:

Sbarrasubmit3_jOVv19M.pdf

Anonymous on 2023-09-23  [id 4004]

(in reply to Anonymous Comment on 2023-09-13 [id 3979])

Let me thank once again the authors for taking into account my suggestions in the current version of the draft; I am now very happy to recommend it for publication.

---

## Editorial Decision

resubmitted